# Integrated Strength and Fundamental Movement Skill Training in Children: A Pilot Study

**DOI:** 10.3390/children7100161

**Published:** 2020-10-03

**Authors:** Fay Grainger, Alison Innerd, Michael Graham, Matthew Wright

**Affiliations:** Department of Science, School of Health and Life Sciences, Teesside University, Middlesbrough TS1 3BX, UK; faygrainger@outlook.com (F.G.); a.innerd@tees.ac.uk (A.I.); Michael.Graham@tees.ac.uk (M.G.)

**Keywords:** physical activity, FMS, strength training, motor competence, health-related fitness

## Abstract

Competence in fundamental movement skills is essential to enable children to be physically active. We investigated the effect of an integrated fundamental movement skill with a strength training intervention on children’s fundamental movement skills. Seventy-two (53% female) 10- to 11-year-old children from three primary schools assented to take part in this study (87% compliance). Schools were randomly allocated to a control (no intervention; n = 21), fundamental movement skill (FMS) (n = 18) or FMS and strength (FMS^+^; n = 20) group. Interventions were delivered twice weekly for four weeks, in addition to normal physical education. FMS competence was measured through the Canadian agility and movement skills assessment (CAMSA) (product-process) and through countermovement jump (CMJ) and 40-m sprint tests (product). Improvements were observed in the CAMSA in both FMS (4.6, 95% confidence intervals 2.9 to 6.4 Arbitrary Units (AUs), second-generation *p*-value (*p*_δ_) = 0.03) and FMS^+^ (3.9, 2.1 to 5.3 AU, *p*_δ_ = 0.28) with no difference beyond our minimum threshold of 3 AU observed between these intervention groups (*p*_δ_ = 1). Clear improvements in CMJ were observed in FMS^+^ relative to the control (25, 18 to 32%, *p*_δ_ = 0) and FMS groups (15, 6.1 to 24%, *p*_δ_ = 0). These preliminary data suggest combined FMS and strength training warrants further investigation as a tool to develop fundamental movement skills in children.

## 1. Introduction

Physical activity is effective in preventing major adverse health outcomes, such as heart disease, stroke, diabetes and colon and breast cancers [1]. We know that higher levels of physical activity in childhood and adolescence contribute to weight management, improved cardiometabolic parameters, neuromuscular development [2] and mental health [3], as well as positive associations to adult physical activity levels [4,5]. Children aged 5 to 18 years old are advised to aim for an average of at least 60 min per day across the week of moderate to vigorous physical activity (MVPA) [6]. In addition, children are advised to engage in a variety of activities that develop movement skills, muscular fitness and bone strength [6]. However, physical inactivity is a public health concern as only 49% of year 5 and year 6 primary school children in the United Kingdom (UK) were found to achieve the UK physical activity guidelines from 2018 [7]. This may be further impacted by deprivation, as people living in the most deprived areas face significant health inequalities [7]. In addition to the MVPA guidelines, the Chief Medical Officers’ recommendations advise children to engage in activities that develop movement skills, muscular fitness and bone strength [6]. We also know that physical activity levels start to decline from an early age (7–8 years) [8], and it has been argued that enhancing a child’s motor performance could be pivotal to attenuating this decline [5].

Motor performance refers to the interaction between fundamental movement skill competence (actual and perceived) and health-related fitness, conceptualised by Stodden et al. [5] as forming a reciprocal relationship with physical activity across the lifespan [9,10]. Fundamental movement skills are considered the building blocks to more complex, sport-specific skills [11] which do not develop naturally over time [10] and are often categorized into three categories: locomotion (e.g., running, hopping, and skipping), object control (e.g., throwing, catching and kicking) and stability (e.g., landing, turning and bracing). Evidence indicates that typically developing children should be able to master most if not all fundamental movement skills by 6 years of age [12]. However, it has been shown that as little as 11% of 12- to 13-year-olds demonstrate mastery or near mastery of fundamental movement skills [13]. This could be, in part, due to the delivery of school-based physical education which has evolved into a health-centred model with a primary focus on increasing MVPA levels and health education/promotion, with the UK curriculum aiming to ensure all pupils are active for a sustained period of time and lead healthy and active lives [14,15]. An undesirable consequence of this approach is that whilst children may be sufficiently active during physical education classes, the opportunity for children to develop their motor skills may be overlooked, thus potentially reducing their ability to maintain physical activity when leaving school and transitioning into adulthood [5]. Additionally, in a primary school setting, physical education is often not delivered by teachers with specialist physical education teacher training as recommended [16] and a lack of teachers’ confidence in delivering physical activity sessions has been reported by Ofsted [17] and in previous research [18]. This lack of confidence may prevent teachers from delivering activities that require specialist coaching and instruction, such as fundamental movement skills [19]. 

Previous fundamental movement skill intervention studies have shown promising outcomes on MVPA levels in children [19,20,21,22,23]. In contrast, some studies have demonstrated that fundamental movement skill interventions are unsuccessful at improving physical activity levels in children [24,25,26,27]. Robinson and Goodway [28] infer that children do not drastically improve their fundamental movement skill competence through free play; improvements are only seen when in an instructed environment and when progressions are apparent. Furthermore, an individual’s fundamental movement skill competence is dependent on a number of genetic, morphological and metabolic factors [29]. Fundamental movement skill interventions with less favourable outcomes may have lacked the necessary focus on health-related fitness such as cardiovascular fitness or muscular strength, a mediator of the relationship between fundamental movement skill competence and physical activity [5,30,31]. Improvements in certain fundamental movement skills (e.g., long jump, sprint speed, medicine ball toss, and jump height) have been observed in both children and adolescents following muscular strength improvements [32]. Indeed, strength training elicits specific neural adaptations that may enhance motor learning [33,34] and the concept of combining strength and co-ordination training has recently been introduced to the strength and conditioning literature [35]. Thus, a novel approach to attenuating the decline in physical activity in children could be the development of interventions that combine fundamental movement skills and strength training. 

Strength and conditioning-based exercises are safe and effective modalities to enhance physical fitness [36] and reduce injury risk [37] in youth. Improved strength at an earlier age may also provide the skills and confidence required to remain physically active as an adult [38]. Enhanced muscular strength means an individual will have improved health and physical performance, be at lower risk of obesity and have a reduced risk of a sport-related injury [39]. Traditionally, the development of muscular strength is reserved for the aspiring young athlete, though recently it has been argued that physical education teachers should implement a systematic approach to long-term athletic development regardless of age, ability and aspirations [39]. Strength training as a school-based intervention is relatively new; however, recent research has shown that resistance training can have immediate and sustained improvements in upper body muscular fitness and resistance training skill competency in secondary school-aged children [40] and, for shorter duration interventions, markers of functional strength can be improved but not in cardiovascular fitness [41]. Indeed, muscular strength in male adolescents has been associated with a reduction in premature death [42], whilst strength training itself has been shown to positively affect indicators of self-efficacy and global and physical self-worth in 10- to 16-year-old children [43]. However, it has been suggested that there is an increased incidence of childhood dynapenia [44] resulting in further disengagement with physically active pursuits and an increased likelihood of adverse health outcomes later in life [45]. Research exploring the effect of strength training on fundamental movement skill competence in children is still in its infancy. A recent review highlighted the positive impact resistance strength training has on indicators of fundamental movement skill (sprint, squat, jumping and throwing) in youth [46]. The included studies tended to rely on resistance machines to develop strength that are not ordinarily available in a school-based setting and to date there is little research investigating the effects of combining strength and fundamental movement skill interventions in primary schools. Integrating fundamental movement skills and age-appropriate strength training could provide an ideal stimulus for children, but the development of such an intervention requires appropriate pilot and feasibility studies [47]. 

A further limitation of the current literature outlined by Collins et al. [46] is the universal use of product-oriented assessments of fundamental movement skills (the outcome of how high or far a child can jump, for example). Whilst these assessments are valuable in providing a controlled assessment of fundamental movement skills, they may not be sensitive enough to quantify the relationship between movement skill proficiency—inclusive of movement fluency, rhythm and timing—and secondary outcomes such as physical activity levels and strength. Further, product-oriented assessments may be influenced by maturation [39] and are also used as outcome measures for health-related fitness (e.g., EUROFIT, A-CLASS [48,49]). Whilst it is more common in the wider literature for fundamental movements skills to be assessed using process-orientated assessments [50], it has been argued that a combined product and process assessment should be considered as these take into account the transferability of skills to complex activities [51] and so more recent holistic assessments of movement competency have been developed [51,52]. For example, these are dynamic assessments which test the ability to fluently combine locomotive (speed and agility), object control and balance and stabilization skills [51,52]. Therefore, the aim of this initial, exploratory pilot study was to assess the effect of fundamental movement skill training alone, or in combination with strength training, on fundamental movement skills, measured using a combined product-process assessment and product-only assessments.

## 2. Materials and Methods

### 2.1. Participants

Following ethical clearance (School of Social Sciences and Law Ethics Committee SSSBLRECSTUD1449) and Head Teacher and parental consent, 72 (53% female) 10- to 11-year-old children from three primary schools assented to take part in this study. The primary schools recruited were from an area in the north east of England classified as being in the lowest 10% on the Index of Multiple Deprivation (English indices of deprivation: Department for Communities and Local Government). Children from areas of high deprivation exhibit less developed movement skills than those from more affluent areas [53] and may require additional movement development interventions [16]. Schools were identified from the same local authority (in the UK a local authority is an organisation officially responsible for all public services in a particular geographical area), who followed the same curriculum, and all eligible schools were invited to participate; the first three responders were recruited. Schools were randomly allocated, using the names out of a hat method, to either a control group (no intervention), a fundamental movement skill group (FMS) or a fundamental movement skill group and strength (FMS^+^) group. Poor attendance in the intervention groups (<85% attendance), absent at time of post-test, injury at time of post-test and incomplete baseline data resulted in 13 participants being excluded from the analysis (Figure 1). Characteristics for the remaining 59 participants can be seen in Table 1. The protocol was registered retrospectively on clinicaltrials.gov (trial number NCT04458844) in July 2020.

### 2.2. Procedures

Baseline data were collected mid-June 2018 at the primary schools. The order of testing performed was counterbalanced to remove the order effect on outcome measures. We were unable to standardise rest periods for each child, but the testing was conducted so that children had approximately 10 min recovery between measures. The children completed the post-test battery in the same order that they completed the tests at baseline. Body mass and stature were measured to the nearest 0.1 cm and 0.1 kg, respectively (Seca Medical measurement systems and scales, Birmingham, UK). Following baseline measures, participants in the intervention groups completed a 4-week intervention, respectively, with outcome measures taken the week following the final session (Figure 1).

### 2.3. Intervention 

Intervention programmes were designed based on previous research [54,55] to ensure all exercises were developmentally appropriate for the age of the participants. No formal feedback was received from teachers or children prior to or throughout the intervention; however, informal discussions, authors’ previous experience and ongoing reflective dialog were used to ensure the delivery was appropriate. Each session consisted of five activities and lasted 50–60 min. Approximately half of the activity was identical for the FMS and FMS^+^ groups, while the other half differed, focusing on skill development for the FMS group and strength development for the FMS^+^ group (Table 2 and Table 3). The FMS and FMS^+^ group received their sessions twice a week for four weeks. The sessions were delivered at least 48 h apart to allow recovery and to reduce the risk of fatigue affecting performance [56]. All sessions were led by F.G., with the assistance of a qualified sports coach. The focus on coaching was important to ensure that movement skills were taught, and strength exercises were executed with technical proficiency. As such the pedagogical approach to delivery included direct instruction, particularly in the early stages and for strength exercises (squat or lunge). However, the coaching of these sessions was fluent and influenced by the approach outlined by Wright and Lass [57] for pre-pubescent athletes. This included a number of principles of non-linear pedagogy such as an external focus of attention and student–coach interaction; e.g., “move like a frog” and providing references for progressions and regressions of exercises. Coaches were trained to use the STEP model by manipulating space, task, equipment and people. The STEP model is endorsed by UK Coaching as an inclusive coaching principle to use with participants to make activities easier or harder. All sessions were in addition to the participants’ usual physical education lesson.

### 2.4. Outcome Measures

The Canadian agility and movement skills assessment (CAMSA), which combines product and process-oriented outcomes was used to assess fundamental movement skills. Full details of the CAMSA can be found in the Canadian Assessment of the Physical Literacy Manual, 2017. The CAMSA was originally developed to offer a more dynamic assessment of children’s fundamental movement skill competence whilst measuring their capability for more complex movements, such as agility. Face validity (using the Delphi process), concurrent validity, and inter-rater and intra-rater reliability have been previously established [52,58,59]. Briefly, the CAMSA requires participants to travel a total distance of 20 m while completing seven different movement skill tasks [52] around an agility-style course. The individual movement skills assessed were (1) 2-footed jumping in and out of three hoops on the ground, (2) sliding from side to side over a 3 m distance, (3) catching a ball and then (4) throwing the ball at a wall target 5 m away, (5) skipping for 5 m, (6) 1-footed hopping in and out of six hoops on the ground, and (7) kicking a soccer ball between two cones placed 5 m away [52]. Process-oriented criteria were used to assess the performance of each fundamental movement skill. In addition, the throw and kick were scored as either hitting or missing a target 5 m away (product assessment). Finally, the overall time taken to complete the course was recorded (product assessment) and converted into a timed score using pre-defined time criteria scores. The CAMSA raw score was calculated by adding the individual skill scores (max of 14 points) and the time criteria score (max of 14 points), giving an individual score between 1 and 28 points. The higher the CAMSA score, the higher the level of fundamental movement skill competence. The CAMSA was demonstrated twice to each group of children with emphasis placed on both effort and speed [52] using standardised scripts. Children then completed two practice trials followed by two timed and scored trials. Verbal cues were given throughout the practice as described in the protocol. During the test trials, verbal cues were used only to remind children of the next task to be performed [52].

Product-oriented assessments of locomotor performance were performed through countermovement jump and sprint tests and strength through grip dynamometry. The choice of testing was made from evaluating previous literature [46,48,49] and determining which test were feasible within our school environments. Sprinting, countermovement jumps and grip strength have shown to be reliable [60,61]. Children followed a set of standardised instructions for all tests. Jump height was calculated from flight time measured by the Optojump Next (Microgate, Italy). Children were allowed up to three trial repetitions before completing three maximal jumps with the best performance used for analysis. Jumps were not counted if the children tucked their legs in the air or landed outside of the 1 m square testing area. Sprint times were recorded over a distance of 40 m to the nearest 0.01 s using an infrared timing gate system (Brower timing systems, Draper, USA). Sprints were repeated three times and the best score was taken for analysis. Grip strength was measured using a handheld dynamometer (Takei Grip-D) and countermovement jump (Optojump Next, Microgate, Italy) were completed three times, with the average score recorded for each test as measures of upper and lower body strength, respectively. 

### 2.5. Statistical Analysis 

We visualised our raw data and inspected the Q-Q plots of the residuals for normality using Jamovi 1.0.5. The distribution was slightly skewed at baseline in the control group for grip strength (left) and 40-m run and, in the FMS + group for grip strength (right), which were overcome after log-transformation, as recommended for physical fitness data [62]. All other data were approximately normal at baseline, confirmed via a Shapiro–Wilk test (*p*-value range: 0.078 to 0.97). Physical fitness outcomes were log-transformed prior to analysis and subsequently back-transformed to derive a percentage change. The reliability of our measures was estimated from the control group data. This provides an estimation of the typical variability in these measures over the time period of the intervention, which is recommended and may better inform sample size calculations for future studies [63]. The change in the mean, inter-class correlation coefficients (3,1), and typical error were calculated. General linear modelling (SPSS Statistics version 24) was chosen to analyse the differences between groups through an analysis of covariance (ANCOVA) with the baseline score as a covariate. 

Uncertainty in all estimates was expressed as 95% confidence intervals (CIs). For the CAMSA, we chose a minimum practically important difference of 3 AU, as this magnitude represents the shift required for a young person to be typically closer or equal (e.g., halfway) to the next competence category [64]. We were unaware of any robust anchor for a minimal-important difference in our physical fitness tests that could be associated with improved health or physical activity in children; thus, we defaulted to a distribution approach here where a substantial difference was estimated as 0.2 × pooled pre-test between-participant standard deviation [65,66]. Standardised (Cohen’s) effect magnitudes were not calculated here partly because they may lack practical context but also as they maybe more vulnerable to sample variance [67]. Finally, to allow comparison of the magnitude of the effects in relation to the minimal-important difference we calculated a second-generation *p*-value (*p*) which describes the proportion of CIs that lie within the minimal-important threshold [68]. Here *p* = 0 and *p* = 1 describe conclusive observations, *p* = 0 indicates that the entire confidence interval lies beyond the minimal threshold while *p* = 1 indicates that the whole confidence interval lies within this threshold and thus is a practically equivalent observation. Additionally, *p* = 0.5 describes data that are strictly inconclusive. 

## 3. Results

Mean and standard deviations for our outcome measures are presented in Table 4 and the reliability of our measures in Table 5. No systematic changes were seen in the CAMSA across the control group although counter movement jump height clearly reduced and grip strength (left) clearly increased. The pre- to post-test inter-class correlation suggests these data were only moderately reliable yet the typical error for this test was below the minimal important difference of 3.0 (AU). Typical error in our physical fitness tests ranged from 5.5 to 12%. 

The between group differences in our outcome measures, 95% confidence intervals and second-generation *p*-values are presented in Figure 2. Improvements were observed in the CAMSA in both intervention groups relative to the control. These improvements were not conclusively or practically meaningful with ~3% of the 95% confidence intervals overlapping the minimal-important thresholds (−3 to +3 AU) in the FMS (*p* = 0.03) and 28% in the FMS^+^ group (*p* = 0.28). The differences between intervention groups were not practically meaningful (*p* = 1), suggesting equivalence [69]. 

We observed clear practical meaningful improvements in countermovement jump height in the FMS^+^ group relative to both the control and FMS groups (*p* = 0). We also observed improvements in the 40 m sprint time for the FMS^+^ when compared to the FMS group but these data were not conclusive (*p* = 0.07). The effects of the interventions on grip strength were clearly inconclusive with the exception of right grip strength where a potential decrease in strength was observed in the FMS group (*p* = 0.36).

## 4. Discussion

A key observation of this study was that a short duration fundamental movement skill intervention appeared to benefit primary school children’s fundamental movement skill competence when measured through a product-process assessment, although this was not conclusive (*p* = 0.03). However, replacing part of this intervention with strength training was just as beneficial but with further benefits observed for maximal locomotor performance when measured through product assessments. For example, the FMS^+^ group also improved in countermovement jump and potentially sprint performance in comparison to the control and FMS groups. This finding supports previous research showing small beneficial effects of resistance training on sprint (95% CI for Hedge’s *g* 0.02–5.7) and vertical jump performance (95% CI for Hedge’s *g* 2.5–5.6) in school-aged children [46]. Our findings suggest that the integration of strength training provides greater benefits beyond that of more typical fundamental movement skill training, at least for product-orientated assessments, and warrants further investigation.

In their systematic review and meta-analysis, Collins et al. [46] showed strength training had clear beneficial effects on a product-oriented assessment of fundamental movement skills. A novel aspect of our study was the inclusion of dynamic assessment of product and process of fundamental movement skills. The CAMSA assesses all three categories of fundamental movements skills in combination, thus providing a valid measure of fundamental movement skill [52]. Given that the proposed reciprocal relationship between fundamental movement skill competence and physical activity appears to be mediated by health-related fitness [5,30], our observed improvements in CAMSA relative to the control are encouraging but not conclusive. Replacing a proportion of fundamental movement skills with strength training resulted in equivalent results. This may be due to a relative benefit in lower limb muscular fitness, indicated by the countermovement jump offsetting a potential decrease in the development of movement competence; however, further research is necessary to confirm this. Future studies may wish to also include appropriate process-orientated assessments of fundamental movement skills.

Typically, strength training interventions have utilised purely resistance strength training. A novel aspect of our study was the integration of strength training within a fundamental movement skill intervention. Flannagan et al. [70] compared traditional machine-based strength training with body weight movement-based strength (animal crawl variations) in a similar cohort to the current study, finding limited effectiveness in both groups. However, their crawling intervention may not have provided the appropriate lower limb stimulus to transfer strength to locomotor performance. In every session of our intervention we included exercises requiring the extension and flexion of the hip, knee and ankle joints. We also included lunging exercises requiring the co-ordinated separation of limbs. Given adaptations to strength training are predominantly neurological in children [32], it is possible the specific neural adaptations from these exercises may transfer better to jumping and sprinting assessments [33] than machine-based resistance exercises and crawling. An alternative categorising of fundamental movement skills suggests seven movements: locomotive, squat, hinge, anti-rotation, rotation, push and pull [71], most of which were either implicitly or explicitly included within our intervention. Interestingly, a high-intensity circuit training intervention over a four-week period was successful in improving strength, but not cardiovascular fitness [41]. The intervention included squatting and jumping (skipping), pushing and lunging activities and improvements in strength, which were postulated to be driven by improved neuromuscular function. Previous research has suggested that Olympic weightlifting training, which combines most of these movements, can enhance muscular co-ordination specific to jumping (i.e., timing of extension of limbs) and landing (co-contraction of musculature) [72]. This provides further support to the belief that the specificity of the training type is related to the nature of the neurological adaptations observed. Therefore, we feel there is scope for further investigation of strength training exercises that have biomechanical transfers to fundamental movement skills.

Fundamental movement skills do not necessarily develop naturally [28] and appropriate teaching or manipulation of environments is an important part of developing these skills [73]. Furthermore, in adolescents, Wright et al. [74] emphasise the importance of coaching exercise to enhance movement quality and central nervous system adaptations given that mentally stimulating exercise may enhance neurogenesis [34,75]. Whilst further research is needed to understand the ideal pedagogical approach [16], a potential limitation of the current study was that the intervention was largely structured and researcher-led. Longer term interventions may provide a greater scope for co-design with teachers and children, potentially enhancing creativity, ownership and integration to the curriculum. Teacher-led resistance training interventions have shown improvements in resistance training skill and upper body muscular fitness [40]. Innovative research in pre-school children has demonstrates the benefits of storytelling on fundamental movement skill interventions [76] and creative solutions could be explored with older children with the purpose of providing a further adaptative stimulus for strength development.

The reliability of the CAMSA over the intervention period was acceptable, the upper 95% CI for the typical error was below our minimal-practical important difference and our inter-class correlation coefficients were acceptable but lower than those reported previously [52,59]. Inter-class correlation coefficients for product-based fundamental movement skills tests and grip strength were good to excellent; however, there was some noticeable variation which should be considered if researchers wish to use these measures over similar intervention periods (see Table 2). Systematic changes in performance in the control group were generally trivial; however, countermovement jump height decreased substantially. Therefore, it is important to note that the beneficial effects observed for our intervention for jump performance represented an attenuation of the decline in performance in the control group. A limitation of this small-scale cluster randomised control trial was the use of only one school for each condition. Additionally, the decrement may have been observed in either of our other schools if they had been allocated to the control condition. It is also worth noting the clear benefits of the additional strength training on sprint performance when compared to fundamental movement skill alone, which was somewhat offset by the possible harmful effects observed in the FMS group. We cannot discount environmental or other factors such as time-tabling differences, or the potential effect of “special days” (away trips or sports days) that may have affected the results in one particular school but not another [47]. A larger trial with multiple clusters and measurement would enable estimations of inter-cluster variation and inter-cluster correlation coefficients to help inform an appropriate sample size for a fully powered trial. However, this is not justifiable in the early stage of a complex intervention, which requires appropriate pilot and feasibility studies to ensure all elements of intervention, study design, outcome measures and implementation are in place first [47]. Our data provide a positive first step in the development of a combined fundamental movement skill and strength training approach to the development fundamental movement skill competence in schools, and despite decrements in jump performance in the control, our FMS and strength intervention had clear benefits when compared to the FMS-only group. Therefore, this supports the notion that a combined approach is worthy of further investigation. However, our intervention was intensive in nature and research-led, which does not consider either sustainability or scalability. Working with the teachers and children to co-design and build capacity to embed interventions was beyond the scope of this pilot study but should be a focus of further feasibility projects.

The intervention was delivered over a short-term period (4 weeks), practically, this allowed all data collection and intervention delivery to be complete in one half-term; however, to make sustained changes to fundamental movement skill competence and physical activity participatory levels, longer interventions may be needed. Furthermore, a longer-term follow-up would be recommended for future trials to evaluate more permanent adaptations in strength and fundamental movement skill competence. In the current pilot study, the academic year finished directly after final data collection; therefore, we were unable to include a follow-up or perform any qualitative work exploring the children’s perceptions of the interventions, or the feasibility and acceptability by the schools. However, the intervention compliance was good (FMS^+^ group, 87%; FMS group, 85%; control group, 75%) and similar to previous studies using resistance training, suggesting the children may have enjoyed these sessions [46]. Chronic adaptations are of interest given the complex relationship between fundamental movement skills competence, movement confidence and physical activity [45]. As such, these data should be viewed as exploratory, but the clear effects observed do provide a strong rationale for further research to explore integrated FMS and strength in primary school settings.

## 5. Conclusions

In conclusion, this exploratory pilot study observed potential positive effects for the integrated combination of fundamental movement skill and strength training when delivered in a primary school setting. These benefits were observed with both product and product-process assessment of fundamental movement skill. Further research is warranted to evaluate the effectiveness of this intervention type in larger scale trials across multiple clusters and over longer periods of time (both intervention and follow-up).

## Figures and Tables

**Figure 1 children-07-00161-f001:**
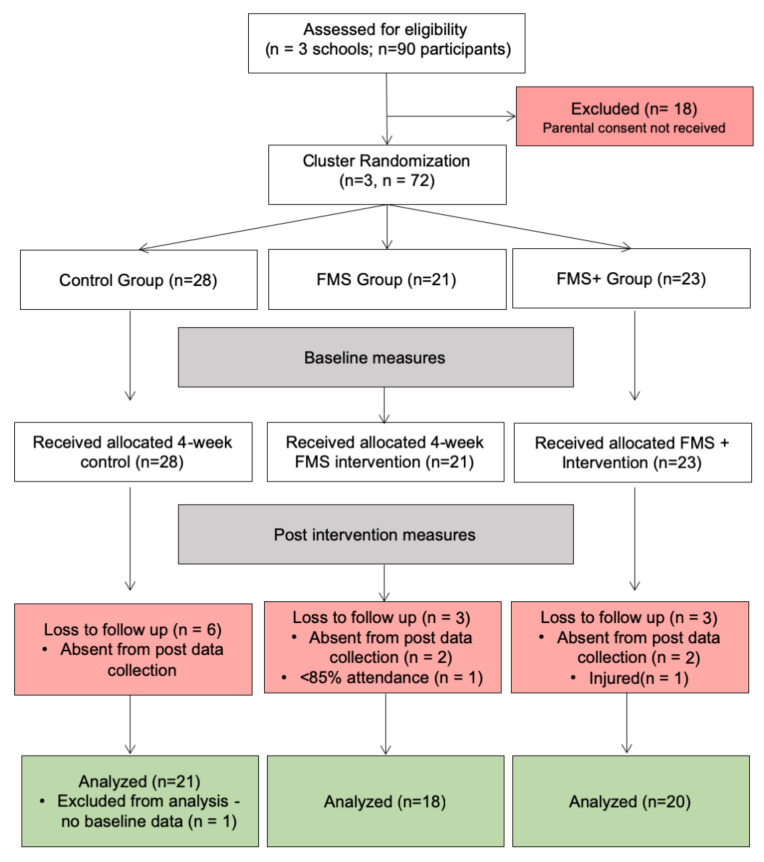
CONSORT flow diagram illustrating the recruitment of participants and those excluded or lost to follow-up.

**Figure 2 children-07-00161-f002:**
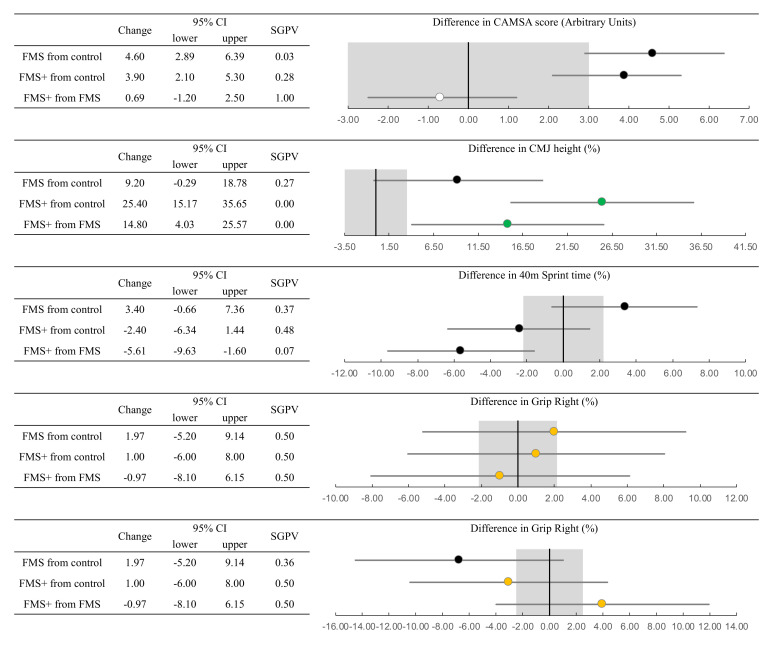
Between-group differences in each outcome measure with 95% confidence intervals (CIs) and second-generation *p*-value (SGPV). Grey shaded area represents the threshold for a minimal-important difference. Markers highlighted in green represent “conclusive” observations (when the entire 95% CI lies beyond the minimal-important threshold. White markers highlight equivalent observations and amber represent effects that are strictly inconclusive.

**Table 1 children-07-00161-t001:** Baseline characteristic data for the children included in each group.

Variable	CON (n = 21; 48% F)	FMS (n = 18; 50% F)	FMS^+^ (n = 20; 60% F)
Stature (cm)	140.8 ± 7.3	147.6 ± 7.0	147.9 ± 5.8
Mass (kg)	37.2 ± 7.8	44.7 ± 10.9	38.9 ± 6.7
Age (years)	10.4 ± 0.3	11.3 ± 0.3	11.2 ± 0.2

Control (CON), Fundamental movement skill group (FMS), Fundamental movement skill and strength group (FMS^+^), female (F).

**Table 2 children-07-00161-t002:** Fundamental movement skill training programme.

	Warm-Up (10 Min)	Activity 2 (10 Min)	Activity 3 (10 Min)	Activity 4 (15 Min)	Activity 5 (15 Min)
Week 1	Inchworm (2 × 15 s) Froggies (2 × 15 s) Bear Crawl (2 × 15 s)	Spots: Locomotor patterns (jump, hop, skip) on a whistle return to “the spot” and balance	Skipping with ropes Over—Under hurdle Ball roll and sprint	Throwing + catch to a wall. Progression—Distance from wall or one-hand catch	Football dribble (gates)
Week 2	Crab walk (2 × 30 s) Spiderman (2 × 30 s) Flamingo(×30 EL)	Balance stuck in the mud (single leg stand—Freed by high 5)	Skipping with ropes Over—Under hurdle Ball roll and sprint	Throw “over the river” to a partner. Progress under to over arm throw. Two to one-hand catch	Dribble around cones and shoot for goal
Week 3	Bunny hops (2 × 15 s) Inchworm (2 × 15 s) Bear Crawl (3 × 30 s)	Balance stuck in the mud (single leg stand—Freed by high 5)	Ladder steps Side-ways over—Under Ball roll and sprint	As above with different size balls	Passing a football in groups of 4 Dribble around cones and pass back
Week 4	Froggies (2 × 15 s) Spiderman (2 × 15 s) Bear Crawl (3 × 30 s)	Bean bag throw and catch (standing on one leg)	Ladder steps Side-ways over—Under Ball roll and sprint	Throwing + catch to a wall. Hoop throws Progression—As above	Passing a football in groups of 4 (introduce defender) Dribble around cones and shoot for goal on command

**Table 3 children-07-00161-t003:** Fundamental movement skill and strength training programme.

	Warm-Up (10 Min)	Activity 2 (10 Min)	Activity 3 (10 Min)	Activity 4 (15 Min)	Activity 5 (15 Min)
Week 1	Inchworm (2 × 15 s) Froggies (2 × 15 s) Bear Crawl (2 × 15 s)	Jumping and landing games (resistance band)	Skipping with ropes Over—Under hurdle Ball roll and sprint	Body weight squat 4 × 10 Squat ball throws with jump	Lung walks with leg drive 4 × 10 Football dribble (gates)
Week 2	Crab walk (2 × 30 s) Spiderman (2 × 30 s) Flamingo (×30 EL)	Jumping and landing games (resistance band)	Skipping with ropes Over—Under hurdle Ball roll and sprint	Body weight squat 4 × 10 Squat ball throws with jump (beanbag to self)	Hip raises on beaches 2 × 10 each leg Dribble around cones and shoot for goal
Week 3	Bunny hops (2 × 15 s) *Inchworm* (2 × 15 s) Bear Crawl (3 × 30 s)	Hopping into hoops Low hurdle jumps Stuck in the Mud	Ladder steps Side-ways over—Under Ball roll and sprint	Squat (1 kg med ball) 4 × 10 Squat ball throws with jump (1 kg)	Press ups 4 × 10 Dribble around cones and pass back
Week 4	Froggies (2 × 15 s) Spiderman (2 × 15 s) Bear Crawl (3 × 30 s)	Hopping into hoops Low hurdle jumps Stuck in the Mud	Ladder steps Side-ways over—Under Ball roll and sprint	Squat (2 kg med ball) 4 × 10 Squat ball throws with jump (1 kg)	Lung walks with leg drive 4 × 10 Dribble around cones and shoot for goal on command

**Table 4 children-07-00161-t004:** Baseline and post-intervention mean and standard deviation for each group.

Outcome Measure		Control	FMS	FMS^+^
CAMSA (AU)	Baseline	18.1 ± 4.2	14.7 ± 2.7	19.2 ± 4.3
	Post	18.0 ± 3.5	20.6 ± 2.5	22.2 ± 3.5
CMJ (cm)	Baseline	18.9 ± 5.8	17.1 ± 4.6	22.3 ± 4.7
	Post	15.6 ± 4.5	15.9 ± 3.6	22.4 ± 4.8
40 m Time (s)	Baseline	6.3 ± 1.0	6.3 ± 0.5	6.0 ± 0.6
	Post	6.3 ± 0.8	6.6 ± 0.7	5.9 ± 0.7
Grip L (kg)	Baseline	13.0 ± 3.1	16.4 ± 3.2	14.9 ± 4.2
	Post	13.7 ± 2.8	15.8 ± 3.1	15.2 ± 4.7
Grip R (kg)	Baseline	13.0 ± 3.1	16.4 ± 3.4	16.1 ± 4.3
	Post	13.7 ± 2.8	16.7 ± 3.0	16.4 ± 4.9

Canadian agility and movement skills assessment (CAMSA), Arbitrary units (AU), Counter movement jump (CMJ).

**Table 5 children-07-00161-t005:** Variation in outcome measures for our control group.

	Change in the Mean (95% CI)	Typical Error (95% CI)	ICC_3,1_ (95% CI)
CAMSA (AU)	0.0 (−1.3, 1.3)	2.0 (1.6, 2.9)	0.73 (0.44, 0.88)
CMJ (%)	−16 (−22, −10)	12 (9.4, 18)	0.88 (0.74, 0.95)
40 m Run (%)	0.1 (−3.2, 3.5)	5.5 (4.2, 7.9)	0.86 (0.69, 0.94)
Grip Left (%)	6.1 (1.1, 11)	8.0 (6.1, 12)	0.91 (0.79, 0.96)
Grip Right (%)	2.1 (−2.2, 6.5)	7.1] (5.4, 10)	0.90 (0.76, 0.96)

Key: Confidence interval (CI); Inter-class correlation coefficient (ICC_3,1_).

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
