# Peer review of "Integrated Strength and Fundamental Movement Skill Training in Children: A Pilot Study"

_children, 2020, doi:10.3390/children7100161_

Round 1

Reviewer 1 Report

The manuscript sets out to investigate the differences in FMS and physical performance in young children following one of three interventions (control v FMS v FMS + RT). On the whole, the study is well designed and well conducted. However, I have minor issues with the novelty and significance of the study. I think the rationale needs to be stronger to highlight what this is, and to show what this study is adding to the current evidence base. There are a number of sections that require more detail, and also typographical errors throughout. 

General comments:

The introduction needs to form a better rationale for the study. At the moment, it doesn't really highlight what this study is adding to the evidence base. We know FMS will improve following resistance training. What is novel about this study? While there are some novel statistics, I think a greater rationale is needed for this approach, and why this is used rather than effect sizes. The discussion needs to actually explain the results. This section is rather vague and doesn't really highlight and explain the findings. This needs greater detail and more attention. 

Specific comments:

Abstract:

Line 10 - replace physical with physically

Line 12 - school should be plural

Introduction: 

Line 26 - list some of these here

Lines 26 - 29 - why is this important? Close the loop on this - reduce health issues, mental and physical? 

Line 35 - why is this? And how can we prevent it?

Line 36 - 38 - what is actually being done here at the moment that prevents FMS development?

Line 41 - need to highlight that children need these skills so that they can maintain physical activity when leaving school and transitioning into adulthood. This is the major issue we face!

Line 49 - Be consistent with use of FMS as acronym 

Line 52 - I would make sure this is relevant to youth and cite a better reference than a textbook. A number of systematic reviews have been published recently that highlight these findings.

Line 54 - in youth? 

Lune 56 and 57 - combine sentences to avoid very short sentences 

Line 59 - need more background here before stating this. How can it build confidence? 

Line 62 - this doesn’t fit here. Move to 2nd paragraph of introduction

Line 77 - This seems like key study in this area. Can you expand on this study and form a more detailed rationale from this. 

Line 78 - what are these?

Method:

Line 96 - replace been with being 

Line 120 - How were sessions progressed or regressed? If children couldn't do an exercise, was it coached or modified? This section needs a lot more detail. 

Line 124 - what was happening in PE lessons? Games based? More detail needed

Line 148 - How was cueing during the testing standardised? 

Line 149 - Were they cued during these tests?

Line 150 - spelling - calculated 

Line 156 - How was grip strength assessed?

Line 167 - What reliability was assessed? Within session or between session?

Line 174 - Are you terming this smallest worthwhile change?

Line 177 - Personally, I would like to see effect sizes included within analysis. I think they tell us significantly more than p values. I don’t understand your point that they lack practical context?

Line 179 - first time I have come across this method. Can you justify why it was used? 

Results

Well presented figures and text supports these

Discussion

Line 217 - If I have understood the analysis correctly, in theory, the statistical difference between intervention groups and control is non-significant for the FMS test. Is this correct? You may need to temper this paragraph if this is the case. 

Line 221 - to a greater degree than the other two groups?

Line 234 - avoid personalisation - "…to the current study."

Line 240 - compared to what? Machine based resistance exercise and crawling?

Line 241 - this seems to lack context. Why is olympic lifting training referred to here? What about other resistance training interventions in children that have improved sprint and jump? Unclear on why that comparison is made. 

Line 246 - spelling - enhance 

Line 246 - 254 - Move this paragraph later - doesn’t really add anything to the diccussion - expand on what needs to be measured in future research to give context to this section

Line 255 - 269 - need to relate this to the current study, otherwise this reads as a literature review rather than explaining current findings.

Author Response

Dear Reviewer 1,

Thank you for your time and expertise in reviewing our manuscript. We feel the amendments made in response to comments has substantially improved the manuscript.

Please see attached response to both reviewers comments. 

Reviewer 2 Report

The authors are to be commended on pursing such an important research area. An area of personal and professional research interest and passion to me. 

I have made several comments and suggestions below for the authors to consider, in order to progress this paper to publication. 

Of most importance is to unpack/acknowledge/include and potentially reframe this piece to reflect 'motor performance' research/literature. Particular attention should be paid to the HRF aspects (motor abilities) that have shown to improve motor performance. This could be then used to justify the measures selected for the fitness outcomes.  A recent piece that may provide stimulus is:

Maike Tietjens, Lisa M. Barnett, Dennis Dreiskämper, Benjamin Holfelder, Till Onno Utesch, Natalie Lander, Trina Hinkley & Nadja Schott (2020) Conceptualising and testing the relationship between actual and perceived motor performance: A cross-cultural comparison in children from Australia and Germany, Journal of Sports Sciences, DOI: 10.1080/02640414.2020.1766169 

In addition - I would like to see CAMSA scores separated for analysis and results. What aspect is driving the improvement in these overall scores - was it an increase in skill quality or and increase in speed. This could feed into why and how each intervention arm resulted in certain outcomes.

Furthermore, I would like the inclusion of instructional frameworks/ pedagogical theory. The lesson plans look very much like 'direct instruction' with little to no autonomy for student voice and choice. This contradicts research around effective interventions in this area. If the authors believe this is not the case  - evidence to support the selected sessions needs to be provided.

Finally, this appears to be a very intensive intervention with limited scope for scalability and sustainability. The ultimate aim of behaviour change interventions is for them to be sustained and embedded in routine practice. The current design does not appear to facilitate this (e.g., researcher lead, no co-design nor capacity building with staff), and is therefore a potential limitation. Perhaps worthy of discussion or consideration when scaling up.

Specific Comments to follow

One author is repeated twice in author list

Abstract 

Line 10 - should this read physically active

Line 10 - integrated FMS what? program/intervention  - please add detail

Line 19 - clarify meaning of 'practical'

Line 19 - any change in control group?

Numbers can be removed from key words

Introduction

Line 34: 2011 nearly a decade ago - is there updated sources

Line 41: please clarify why a focus on a health-centred model that focuses on increasing MVPA would jeopardise the development of FMS and fitness. 

Line 42: suggest the inclusion of 'often' categorized' as there are several ways motor skills are and can be categorized. 

Line 44: you state that the national curriculum includes FMS - which seems to contradict your statement from line 41 around omission/decreased focus of FMS  - please clarify 

Line 47: suggest the inclusion of 'typically developing' children 

Line 49: discrepancy between - please clarify what is meant here

Line 49: suggest dynopenia removed from here and introduced after fitness paragraph. Also suggest more detail added here as to the declining levels of fitness.

Line 58: PE curriculums have focused on HRF - perhaps some detail around why there are declining levels even though there is a focus on this in the curriculum.

Line 65-66: please be mindful of the specific aims of these interventions. In all interventions was the aim to improve both FMS and fitness?

Line 69: as mentioned above - I suggest the inclusion of motor performance research/literature here  - particularly around the role of HRF aspects such as strength, speed, endurance and flexibility (motor abilities) in motor performance. This detail will help justify/rationalise the study and the measures. Also suggest consideration of the dynamic nature of the CAMSA - the inclusion of speed/agility and stability within the assessment. What are the implications of this?

Line 77: please clarify what is meant by 'indicators'

Line 79: please explain why product only measures may be a limitation. Also worthy of discussion is the link between fitness based aspects (e.g., strength) and product outcomes.  

Line 92: justify why these specific schools were selected   - rationalise as 'at risk' groups for low levels of FMS proficiency  - could be included in introduction. 

Line 99: great variability in BMI  - is this taken into consideration in analysis and outcomes? 

Line 114: As mentioned there are several questions around the sustainability of such as design in the real world. 4 weeks, twice a week, intensive, researcher led....the long term implementation of this and translation into practice to have sustained effect is then questionable. Aspects to consider:

  • were teachers/stakeholders involved in co-design/capacity building/professional learning?
  • were students included is regards to choice/voice?  

Table 2

What theoretical framework/evidence base or pedagogical model guided these sessions? Detail required.

As mentioned it appears very researcher/lead - direct instruction with low levels or provision of student autonomy. Justification as to the choice of activities and delivery mode/ dose/frequency/methods required. 

Line 145: reliability and validity details for the CAMSA in regards to use in similar populations to this study is required. Also as mentioned - may be worth mentioning the dynamic nature of this instrument that captures more than just object control and locomotor proficiency.  

Control group: please document what the control group did. Perhaps worthy of mentioning how these groups were 'matched' - eg did they have similar amounts of sport and PE and were they covering similar curriculum content.

Line 145-159: more justification/rationalisation around why these specific measures were selected. Also validiy and reliability measures need to be included.

Analysis

Did you account for BMI?

Was there consideration of amount of PE/Sport exposure? 

Results

I suggest looking at the CAMSA results separately i.e. time score and skill score. It would be interesting to identify which aspect was driving the improvement -  particularly considering the focus of your two intervention arms. 

Discussion

As mentioned I suggest looking at motor performance literature. 

Author Response

Dear Reviewer 2,

Thank you for your time and expertise in reviewing our manuscript. We feel the amendments made in response to comments has substantially improved the manuscript.

Please see attached response to both reviewers comments. 

Round 2

Reviewer 2 Report

This paper has been much improved via the revision process. However, there are a few aspects that require more thought and clarity. The primary concern is around the use of terminology. Please clarify use and distinguish between motor skill, motor performance, motor competence, fundamental movement skill. More specifically the difference between motor performance (which is often viewed as a combination of fitness and skill) and other aforementioned terms used. Clarification and consistency is required throughout.

Please see below for other minor points:

Line 61: Please check use of word 'Teach'

Line 62: please add the implication due to the lack of teacher confidence/competence here - what is the impact of this

Line 74: See comment above around use, understanding and consistency of terminology 

Line 112: please clarify how fluency, rhythm and timing is measured - as none of the data collection instruments collects this data

Line 121: please provide a reference here

Line 174: please check 'approached'

Line 177: please spell out, elaborate and reference the importance or effectiveness of STEP  

Line 298: please check accuracy in regards to assessment of all three categories. As the CAMSA does not explicitly measure stability.

Author Response

We would like to thank the reviewer for the time taken to review our manuscript again. We have considered all of the suggestions made and have provided our point-by-point response (attached) and indicated the changes within the manuscript in red text. Where changes have not been made or only partially made, we have given these much thought and consideration providing a justification below. Once again, we feel these comments have improved the manuscript, particularly the consistency in terminology used throughout.
